# Leisure-Time Physical Activity as a Pathway to Sustainable Living: Insights on Health, Environment, and Green Consumerism

**DOI:** 10.3390/ijerph21050618

**Published:** 2024-05-14

**Authors:** Han-Jen Niu, Kuei-Shu Huang, Pao-Yuan Huang, Huey-Fang Ju

**Affiliations:** 1Department of Management Sciences, Tamkang University, New Taipei City 251301, Taiwan; 2Office of Physical Education, Tamkang University, New Taipei City 251301, Taiwan; 138490@mail.tku.edu.tw; 3Department of Hospitality Management, Hsing Wu University, New Taipei City 244012, Taiwan; 096058@mail.hwu.edu.tw; 4Center for Teacher Education, Tamkang University, New Taipei City 251301, Taiwan

**Keywords:** leisure-time physical activity, sports and leisure involvement, green consumerism, sustainable living, health awareness, environmental awareness, eco-responsible consumer approach

## Abstract

In the aftermath of the COVID-19 pandemic, the intricate relationship between health and the environment has emerged with unparalleled significance. This investigation examines the effect of leisure-time physical activity (LTPA) on health and environmental consciousness and its influence on attitudes towards green/sustainable products among 533 individuals. Utilizing linear structural modeling and regression analysis, the findings reveal that participation in sports and leisure activities significantly mediates the connection between individual well-being and eco-responsible consumer behaviors. Highlighting LTPA’s crucial role in enhancing environmental awareness, this study offers invaluable perspectives for the green product sector. It advocates for the development of strategies that align with consumers’ environmental values, underscoring the essential function of sports and leisure in fostering sustainable consumer practices. Crucially, this underscores the intertwined nature of environmental sustainability and individual health, highlighting their mutual dependence.

## 1. Introduction

In the contemporary era, the dual crises of climate change and dwindling natural resources have emerged as paramount concerns, reshaped human lifestyles, and exerted significant impacts on the environment [1]. As societies grapple with the aftermath of the COVID-19 pandemic, the intricate interplay between human health and the environment has come into sharper focus, underscoring the pressing need for a paradigm shift towards sustainable living [2]. Furthermore, discussions surrounding “nature deficit disorder” and its psychological implications have underscored the importance of reconnecting with the natural world to mitigate adverse effects on mental well-being [3].

In this context, leisure-time physical activity (LTPA), as delineated by Steinbach and Graf [4], assumes multifaceted significance. Beyond its well-documented benefits for physical health and mental wellness, LTPA serves as a conduit for fostering a deeper connection with nature [5,6]. Through activities such as hiking, gardening, or simply spending time outdoors, individuals engage directly with their natural surroundings, cultivating a heightened sense of appreciation and awareness of the environment [7]. This experiential interaction with nature not only enhances individual well-being but also nurtures a sense of environmental stewardship and responsibility [8]. As societies navigate the complexities of modern living, integrating LTPA into daily routines emerges as a promising avenue for promoting both personal and environmental health. By encouraging active participation in leisure activities that facilitate direct engagement with nature, policymakers and public health officials can promote holistic well-being while fostering a more sustainable relationship with the environment [9]. Moreover, initiatives aimed at promoting LTPA can serve as catalysts for broader environmental conservation efforts, mobilizing communities to actively participate in safeguarding natural ecosystems for future generations [10].

Acknowledging the intertwined challenges of maintaining public health and ensuring environmental sustainability, this study delves into the nuanced relationships between leisure-time physical activities (LTPA), health awareness, environmental consciousness, and consumer behaviors towards green/sustainable products. Our investigation seeks to unravel the aggregate impact of sports and leisure involvement, along with health and environmental awareness, on green product purchasing behaviors. The existing literature has underscored the positive correlation between engagement in sports and leisure activities and heightened health awareness [7,8]. This heightened awareness, in turn, is posited to foster a greater acceptance of green/sustainable products [9]. Moreover, physical activities have been identified as instrumental in promoting environmental stewardship [11], with individuals actively engaged in leisure pursuits often exhibiting a deeper appreciation for environmental conservation efforts. Despite the recognized roles of LTPA, health awareness, and environmental consciousness, the intricate connections between these variables and green product purchasing decisions remain relatively understudied. Our research endeavors to address this gap, offering valuable insights for industry stakeholders and policymakers alike. By elucidating the pathways through which leisure activities, health awareness, and environmental consciousness influence consumer behaviors, we aim to inform strategic decision-making processes and facilitate the development of targeted interventions aimed at promoting sustainable consumption practices. Through our comprehensive analysis, we seek to contribute to a more nuanced understanding of the complex interplay between individual behaviors, lifestyle choices, and environmental consciousness in the context of green consumerism.

With the growing corporate emphasis on ESG (environmental, social, and governance) imperatives and the surge in consumer demand for green/sustainable products, the role of leisure-time physical activity (LTPA) in elevating health and environmental awareness becomes even more pertinent [7,10,12]. As businesses increasingly recognize the importance of integrating sustainability principles into their operations, understanding the mechanisms through which LTPA influences consumer behavior is paramount. Moreover, health and environmental consciousness are pivotal in shaping consumer purchasing decisions, with individuals exhibiting positive attitudes towards these aspects more inclined towards green purchases [2,13,14,15,16]. This underscores the significance of not only promoting the environmental benefits of sustainable products but also emphasizing their potential health-related advantages. By delving into these variables (sports and leisure involvement, health awareness, environment awareness, green/sustainable product attitude, and purchase intention) and elucidating their interconnectedness, this study aims to deepen understanding of consumer perceptions and choices regarding green/sustainable products.

Through this comprehensive investigation, we endeavor to equip businesses with strategic insights for marketing sustainable products effectively. By highlighting the multifaceted benefits of green consumption, including both environmental and health considerations, we aim to foster consumption patterns that are not only environmentally friendly but also conducive to individual well-being. In doing so, we contribute to the ongoing discourse on sustainable consumerism and offer practical guidance for businesses seeking to align their offerings with evolving consumer preferences and values.

## 2. Literature Review

### 2.1. Sports and Leisure Involvement

The concept of ‘involvement’ within social sciences is pivotal for understanding the depth of an individual’s cognitive and emotional engagement with a particular domain, such as sports and leisure activities [7]. This involvement plays a crucial role in shaping consumer behavior, influencing decisions from brand loyalty to satisfaction in various spheres [8,9], and affecting the processing and retention of information [17].

Building on insights from Steinbach and Graf [4], this section broadens the discussion to leisure-time physical activity (LTPA), which includes any physical activity undertaken during discretionary time. The research underscores the significant health benefits of LTPA, ranging from mitigating cardiovascular disease risks [18,19] to enhancing mental health and overall well-being [20,21]. Moreover, LTPA’s role in reducing stroke incidence [22] and overall mortality [18] underscores its vital importance within the broader context of health and lifestyle choices.

Engagement in outdoor sports and leisure activities represents a nuanced aspect of involvement, specifically encompassing cognitive, emotional, and physical engagement with these pursuits [12]. Such engagement not only promotes physical health and enhances health consciousness but also fosters a deep appreciation for the natural environment [10]. Highly involved individuals often adopt healthier lifestyles, integrating regular physical activities that serve as preventive measures against chronic conditions and psychological distress [23,24].

Additionally, emphasizing the exclusive connection between sports and leisure activities conducted in natural settings can deepen our understanding of the intrinsic relationship between these activities and environmental consciousness. Well-being studies have highlighted the therapeutic benefits of natural healing gardens and mind-body practices, further emphasizing the importance of engaging in sports and leisure activities in natural spaces [25]. Engaged individuals who actively participate in outdoor sports and leisure activities often develop a deeper connection with nature. This enhanced connection not only fosters environmental awareness but also motivates individuals to become advocates for environmental preservation and to adopt eco-friendly practices [11]. Through regular engagement in outdoor activities, individuals reinforce their commitment to safeguarding the environment, thereby contributing to broader environmental conservation efforts [26].

### 2.2. Health Awareness

Health awareness is defined as an individual’s cognition and understanding of health information, potential risks, and their consequent outcomes. As health awareness increases, people tend to make choices that are more beneficial to their health. This concept encompasses not only the recognition of one’s health status but also strategies to promote overall health and well-being through lifestyle choices [27,28]. Delving deeper, health awareness embodies an individual’s health knowledge, attitudes, and behaviors, profoundly influencing and directing their health management strategies as well as methods of disease prevention and treatment [28,29]. More broadly, health awareness is not only related to individual physiological health but also stands as a pivotal factor affecting the overall well-being of society [28].

The aim is to encourage individuals to adopt healthier lifestyles, understand and manage health risks, and seek appropriate medical resources when necessary. The focus is on enabling individuals to participate more actively in their health management and, through effective education and information dissemination, make healthier decisions [2,29]. As individuals enhance their awareness of health risks, they often take more preventative measures, thereby improving their quality of life. Additionally, this contributes to reducing unnecessary medical interventions, subsequently conserving medical resources. As a result, benefits are gained both in terms of quality of life and socio-economic advantages [30].

The relationship between sports, leisure activities, and health awareness can be further elucidated by examining their interaction within natural environments. Research on well-being studies and the benefits of engaging in natural healing gardens and mind–body practices provides valuable insights into this connection. Engaging in sports and leisure activities in natural spaces offers unique opportunities to enhance health awareness through various mechanisms. Firstly, individuals who participate in these activities often report heightened physical perceptions, as they directly experience the benefits of physical activity amidst natural surroundings. Studies by Lin and Niu [25] and Robert, Hinds and Camic [3] highlight the positive impact of natural environments on physical and mental well-being, further emphasizing the importance of conducting sports and leisure activities in natural spaces.

Moreover, social interactions during sports and leisure activities conducted in natural environments play a crucial role in promoting health awareness. Research by Lee et al. [31] indicates that individuals engaged in outdoor activities are more receptive to health information and are more likely to adopt healthy behaviors. Additionally, the exchange of health-related information among participants fosters a supportive environment conducive to enhancing health awareness [3,32]. In light of these perspectives, we propose the following hypothesis, H1:

**H1:** 
*Sports and leisure involvement has a positive effect on health awareness.*


### 2.3. Environmental Awareness

Environmental Awareness elucidates an individual’s and community’s profound understanding of the Earth’s natural environment. This understanding encompasses the workings of ecosystems, human impacts on the environment, and methods to protect and conserve natural resources. Recent studies have further explored the intricate relationships between environmental awareness and social, economic, political, and cultural contexts [15,33]. The concept aims to encourage individuals and communities to adopt ecologically responsible actions. By enhancing sensitivity to environmental issues, individuals are likely to more fervently support sustainable environmental measures.

From both academic and applied perspectives, environmental awareness holds unique importance. On the one hand, it stands at the core of achieving sustainable development; when community members possess adequate environmental awareness, they are more proactive in acting to reduce negative impacts on the planet [34]. On the other hand, by understanding environmental vulnerabilities, people might exercise more caution in resource usage, thereby reducing wastage [13,15]. Moreover, environmental awareness can guide individuals to make more environmentally friendly choices in their daily lives [35]. Additionally, through education and participation, we can elevate public engagement, fostering proactive measures in environmental protection, consensus building, and ensuring sustainable resource utilization [13].

Furthermore, many recent studies have begun to delve deeply into the relationship between sports and leisure involvement and environmental awareness [14]. Through leisure activities conducted in natural settings, participants might become more familiar with and cherish the natural environment, consequently taking more active steps to protect it [5]. Several leisure activities also include environmental education components, such as diving courses emphasizing the importance of coral reefs and protection measures [36]. This kind of education not only strengthens participants’ environmental awareness but also spurs them to exhibit more eco-friendly behaviors in daily life.

Overall, sports and leisure involvement not only offers individuals a healthy lifestyle but also effectively elevates their environmental awareness through hands-on experiences, education, and community interaction. Thus, we hypothesize the following:

**H2:** 
*Sports and leisure involvement has a positive effect on environmental awareness.*


### 2.4. Green/Sustainable Product Attitude

The green/sustainable product attitude can be defined as consumers’ evaluation, emotional response, and behavioral inclination towards products that, during their production, use, and disposal, can reduce adverse environmental impacts [25,32]. This attitude is not only a response to the eco-friendly features of a product but also reflects the understanding of sustainable practices throughout the product’s life cycle. Its research significance lies in uncovering consumers’ cognition, evaluation, and action intentions toward environmentally responsible or sustainable products.

The importance of understanding the green/sustainable product attitude is rooted in its influence on consumer purchasing decisions. A positive attitude might amplify purchasing intentions, further driving market demand and prompting businesses to adopt sustainable production methods [37]. Moreover, formulating effective green marketing and communication strategies targeting environmentally conscious consumers can provide businesses with a competitive edge in the market, while simultaneously advancing the goals of sustainable development [38].

With escalating global environmental issues, consumers are increasingly concerned about global environmental topics, making them more likely to prioritize products with environmental benefits, thereby becoming a method of engaging in environmental protection [39]. Therefore, businesses need to understand and respond to consumers’ green/sustainable product attitudes to enhance their market competitiveness and improve their ESG image.

In today’s society, consumers’ awareness of their own health and environmental well-being continues to deepen. This foundational understanding pushes people to lean towards products perceived to have positive impacts on both health and the environment. Research by Grankvist and Biel [40] and Amberg and Fogarassy [41] suggests that once consumers recognize the environmental benefits of a product, they are more likely to associate it with health benefits, subsequently increasing their purchasing intent for that product. Furthermore, Paul, Modi, and Patel [42] established a positive correlation between consumers’ health consciousness and their intent to purchase green products. Do Paco, Shiel, and Alves [43] also emphasized that consumers, when choosing products, are driven by concerns for both health and the environment. If consumers perceive a product as maintaining both personal health and the environment, they are more likely to purchase it. Based on the aforementioned literature, this study proposes the following hypothesis:

**H3:** 
*Consumers’ health awareness has a positive effect on their attitude toward green/sustainable products.*


In recent years, with the rising prominence of environmental issues, there has been a gradual increase in consumers’ environmental awareness, influencing their purchasing behavior and attitudes towards green products. Research by ElHaffar, Durif, and Dubé [13] and Smith and Brower [44] both suggest that consumers with high environmental consciousness tend to choose products perceived as eco-friendly. This not only aligns with their aspirations to protect the environment but also furthers the realization of their personal values. Additionally, Kautish, and Sharma [45] and Lin and Niu [25] found a distinct positive correlation between consumers’ environmental awareness and their attitude toward green products. As consumers’ concerns for environmental health and sustainability deepen, they become more inclined to choose and support green products that resonate with their values. Furthermore, Sun, Teh, and Linton [46] also highlighted that enhancing environmental education and advocacy can elevate consumer environmental consciousness, positively driving the green product market. Synthesizing the above literature, there is a clear positive association between consumers’ environmental consciousness and their attitudes toward green products. Based on these findings, this study proposes the following hypothesis:

**H4:** 
*Consumers’ environmental awareness has a positive impact on their attitude towards green products.*


Involvement in sports and leisure characterizes consumers’ participation and enthusiasm in leisure sports activities. Once consumers exhibit a high level of involvement in this aspect, their concern for health and the environment often intensifies, potentially leading to heightened support for green products. Studies by Hinds and Camic [3] and Widawska-Stanisz [47] both affirm that consumers engaged in leisure sports frequently regard environmental health as an integral component of their healthy lifestyle. As such, these consumers are more inclined to opt for green products, perceived as beneficial for both health and the environment.

Furthermore, as consumers’ involvement in leisure sports escalates, their purchasing of green products tends to increase. This could be attributed to these consumers valuing their personal health and quality of life more profoundly, recognizing the purchase of green products as one of the strategies to achieve these objectives [47]. Research by Channa, Tariq, Samo, Ghumro, and Qureshi [48] and Patwary [49] further underscored that consumers deeply involved in leisure sports generally place a greater emphasis on environmental preservation, consequently forming more positive attitudes towards green products. Taken together, the existing literature explicitly suggests a positive association between sports and leisure involvement and consumers’ attitudes toward green products. Based on this, the present study posits the following hypothesis:

**H5:** 
*Involvement in sports and leisure has a significant impact on consumers’ green/sustainable products attitudes.*


In contemporary society, health consciousness and environmental awareness are increasingly emphasized and have profoundly influenced consumers’ purchasing decisions and lifestyles [15,33,40,41]. This study also examines how these two forms of consciousness mediate the relationship between leisure sports involvement and attitudes toward green/sustainable products.

Health consciousness describes an individual’s attention and awareness of their health status. As consumers become more health-conscious, they are likely to engage more in leisure sports activities, viewing them as a vital part of maintaining health [50]. Moreover, a heightened sense of health consciousness may predispose consumers to favor green or sustainable products, as these are perceived to be safer, healthier choices [51]. On the other hand, environmental awareness reflects consumers’ recognition and concern for environmental issues. When consumers are more environmentally conscious, they tend to engage more in outdoor sports and other nature-related leisure activities [52]. Furthermore, a strong environmental awareness correlates with positive attitudes toward green or sustainable products.

In summary, both health consciousness and environmental awareness might mediate the relationship between sports and leisure involvement and attitudes towards green/sustainable products. Based on this, the following hypotheses are proposed:

**H6:** 
*Health awareness mediates the relationship between sports and leisure involvement and attitudes toward green/sustainable products.*


**H7:** 
*Environmental awareness mediates the relationship between sports and leisure involvement and attitudes toward green/sustainable products.*


### 2.5. Green/Sustainable Purchase Intention

Purchase intention can be defined as the likelihood or intention of a consumer to plan or intend to buy a particular product or service. In the context of green or sustainable products, green purchase intention refers to a consumer’s intent to buy environmentally friendly products or services [25,53]. Green purchase intention is influenced not only by consumers’ personal beliefs and attitudes but also by external social and cultural factors. Moreover, understanding purchase intention primarily aims to predict and influence consumers’ buying behaviors. For companies, this provides a strategy to modify consumer behavior towards achieving sustainability objectives [25]. The significance of purchase intention lies in its potential to drive market transformations; when consumers choose to purchase green products, they are not only making an eco-friendly choice but also incentivizing companies to create and sell these products. Thus, green purchase intention becomes a crucial driving force for more sustainable production and consumption.

In recent years, the relationship between green purchase attitudes and the intention to buy green products has received widespread academic attention. Multiple studies have suggested that when consumers have a more positive green purchase attitude, their intention to purchase green products is correspondingly enhanced [25,32]. A green purchase attitude refers to a consumer’s positive or negative evaluations and perceptions when buying green products or services. Additionally, when consumers perceive that purchasing green products offers personal or environmental benefits, their purchasing intent becomes even more pronounced [25].

Lin and Niu [25] further highlighted environmental knowledge, environmental awareness, and social norms as key determinants of green purchase attitudes. Enhancing consumers’ awareness of environmental issues and the benefits of green products can elevate their purchase attitudes and intentions. However, ElHaffar et al. [13] noted a gap between green consumption attitudes and actual purchasing behaviors, suggesting that other external factors like price and product availability also influence consumer buying decisions. Interestingly, when the green purchasing attitude of a consumer is exceptionally strong, the impact of these external factors diminishes. Kollmuss and Agyeman [15] found that while many consumers have the intent to purchase green products, their actual behavior might be hindered by a lack of information or the higher prices of green products. Yet, a positive green purchase attitude can motivate consumers to overcome these obstacles. Based on these findings, this study posits the following hypothesis:

**H8:** 
*A positive green/sustainable purchasing attitude will directly influence the intention to purchase green/sustainable products.*


The link between recreational sports involvement and a consumer’s intention to buy green products can be traced back to the consumer’s environmental concerns and values. Consumers, through participating in recreational sports activities like hiking, camping, and other outdoor sports, often come into direct contact with nature. This closeness to nature likely enhances their awareness and the importance given to environmental conservation [3,47]. Therefore, consumers deeply engaged in these recreational activities are likely to have a higher recognition and intention towards purchasing green products. Based on this, this study proposes the following hypothesis:

**H9:** 
*Sports and leisure involvement positively influences the intention to purchase green/sustainable products.*


## 3. Material and Methods

### 3.1. Participants and Sample Profile

This study enrolled adult participants residing in Taiwan, ranging in age from 18 to 70, with sufficient disposable income. A total of 557 questionnaires were disseminated, employing convenience sampling as the primary method for survey distribution. Following the exclusion of 81 incomplete or invalid responses, 533 questionnaires were deemed valid, resulting in a response rate of 95.6%. Gender distribution among valid responses indicated that 61% were female and 39% were male. Regarding age demographics, 32.5% fell within the 21–30 age bracket, 24.8% were under 20 years old, and 15.2% were aged 31–40. Educational attainment predominantly consisted of college-level qualifications, representing 74.9% of respondents, while approximately 72% of participants were unmarried. Furthermore, approximately 73% of respondents reported prior experience in purchasing green sustainable products. Notably, the top three preferred physical activities among participants were jogging, fitness training, and cycling, respectively. The study framework is outlined as follows:

### 3.2. Measures

All variables were measured using the multi-item Likert-type scales. A 6-point Likert scale, ranging from “strongly disagree” to “strongly agree,” was utilized. The scale items used in this study are as follows: Sports and leisure involvement was assessed by using the Zaichkowsky [7] 6-item scale, health awareness was assessed by using Ellen [54], and environmental awareness was evaluated by adopting the Schlegelmilch, Bohlen, and Diamantopoulos [55] 3-item scale. The purchasing intention 3-item scale was measured with the scale of Shariff [56]. Purchasing behavior was also assessed using the Shariff [56] 3-item scale.

## 4. Results

### 4.1. Sample Profile

The sample profile of survey participants with 533 valid questionnaires are summarized as follows: the sample population revealed that 57.3% were below 30 years old; 73.0% have bought green/sustainable products; 92.5% had a bachelor’s or master’s degree, and 78.4% resided in Northern Taiwan.

### 4.2. Reliability Analysis

The means, standard deviations, and reliabilities among the study variables are displayed in Table 1. The Cronbach’s α, as indicated in this table, shows that the internal consistency reliabilities of all the variables measured in this study were quite respectable. All the Cronbach’s α for the variables are up to 0.7, which represents good reliability.

### 4.3. Confirmatory Factor Analysis

#### 4.3.1. Convergent Validity

The Confirmatory Factor Analysis (CFA) is utilized to validate the factor structure identified by exploratory factor analysis and to ensure that observed variables accurately measure latent variables. For assessing the goodness-of-fit index using CFA, both convergent and discriminant validity analyses are imperative. Convergent validity encompasses three elements: factor loading, composite reliability, and average variance extracted (AVE). Bagozzi and Yi [57] posit that standardized factor loadings should range between 0.6 and 0.95. Composite reliability should exceed 0.6, and AVE should surpass 0.5 based on measurement standards. If CFA yields a factor loading above 0.6 for various dimensions, the item is retained as valid. Otherwise, it is excluded. The findings are detailed in Table 2.

#### 4.3.2. Discriminant Validity Analysis

Discriminant validity analysis delves into the distinctions between variables across multiple dimensions. As posited by Fornell and Larcker [58], it is crucial to juxtapose each dimension with the others to assess the discriminant validity of the scales. The diagonal values stretching from the top left to the bottom right represent the AVE of each dimension. Beneath this diagonal, the values depict the correlation coefficients between the dimensions. A critical criterion to note is that the square root of the AVE values should surpass the correlation coefficients. As revealed in Table 3, not all dimensions are entirely correlated with one another, indicating robust discriminant validity.

### 4.4. Structural Equation Modeling Analysis

Structural equation modeling (SEM) was employed to validate the study’s framework, as depicted in Figure 1. Prior to the analysis, it is essential to assess the goodness-of-fit index to determine the study model’s appropriateness for SEM analysis. Within this model, DF = 183, and the *p*-value < 0.001. Such outcomes indicate that this study’s model is apt for SEM analysis. Several researchers have provided goodness-of-fit criteria, with their findings consolidated in Table 4.

In this study, the SEM analysis is employed to delve into the relationships among the variables. Model validation results reveal that while involvement in sports and leisure does not significantly influence green/sustainable purchase intention, other variables show notable correlations (Figure 2).

The results of the analysis, if the hypotheses are valid, are compiled in Table 5. The results show that most hypotheses in the research model are sustained. The sports and leisure involvement cannot directly affect consumers’ attitudes towards green/sustainable products, indicating that health awareness and environmental awareness are essential, complete mediating variables. Therefore, consumers’ involvement in sports and leisure activities can effectively influence their attitudes towards green/sustainable products through health awareness and environmental awareness, leading them to take purchasing actions for such products.

### 4.5. Mediated Effect

Using “health awareness” and “environmental awareness” as mediating variables, this study examines the mediating effects between “sports and leisure involvement” and “green/sustainable products attitudes.” Regression analysis was employed to test the hypotheses H6 and H7. The results are summarized as follows (Table 6):

The results of the mediation analysis revealed a significant indirect effect, suggesting that both health awareness and environmental awareness act as mediators between sports and leisure involvement and attitudes toward green/sustainable products (*p* < 0.05). More specifically, the total effect of sports and leisure involvement (the independent variable) on green/sustainable product attitudes (the dependent variable) was found to be significant.

Nevertheless, upon controlling for the mediating variables, the direct effect of sports and leisure involvement on green/sustainable product attitudes became non-significant (*p* > 0.05), implying a full mediation. Accordingly, Hypothesis 6 (H6), positing that health awareness fully mediates the relationship between sports and leisure involvement and green/sustainable product attitudes, is supported. Similarly, Hypothesis 7 (H7), which proposes that environmental awareness fully mediates the aforementioned relationship, is also confirmed.

## 5. Conclusions and Application

### 5.1. Conclusions

The landscape of global health and environmental challenges, magnified by the COVID-19 pandemic, underscores the vital interconnectedness of our actions, health, and the planet’s well-being. Grounded in the principles of time geography and the profound benefits of leisure-time physical activity (LTPA) as outlined by Steinbach and Graf [4], this research highlights the pivotal role of engaging in sports and leisure activities not just for personal health improvement but as a catalyst for environmental consciousness. The evidence provided by Holtermann et al. [18], Warburton et al. [19], and others, underscores LTPA’s contribution to mitigating major health risks and enhancing mental well-being, thereby establishing a direct link to eco-responsible behaviors.

The conclusion of this study demonstrates the significant influence of leisure-time physical activity (sports and leisure involvement) on green/sustainable purchase intention (H9), indicating that individuals engaged in leisure-time physical activity are more likely to lean towards purchasing environmentally friendly and sustainable products. However, for green/sustainable purchase attitude (H5), leisure-time physical activity did not show a significant impact. This suggests that while individuals participating in leisure-time physical activity may consider purchasing environmentally friendly and sustainable products, their attitudes towards these products may not necessarily change as a result.

It is noteworthy that through the mediating variables of health awareness and environmental awareness, leisure-time physical activity significantly influences green/sustainable purchase attitudes (H6, H7). This indicates that health and environmental awareness play a crucial role in the influence of leisure-time physical activity on attitudes towards green and sustainable products. Specifically, leisure-time physical activity may increase individuals’ concerns about health and environmental issues, thereby affecting their attitudes towards green and sustainable products.

The insights of this study lie in revealing the significant role of leisure-time physical activity in forging the link between health and environmental awareness. With growing societal concern about environmental issues and health crises, a deeper understanding of the psychological and behavioral foundations influencing environmentally responsible consumer behavior becomes crucial. Thus, the findings of this study offer valuable insights to better understand the motivations and influencing factors behind individual consumer behavior.

This intricate dynamic presents a compelling case for businesses, especially within the green product sector, to leverage this heightened awareness by promoting practices that align with consumers’ growing environmental and health consciousness. This study advocates for a strategic pivot where the promotion of green products is integrated with initiatives aimed at enhancing LTPA, thereby fostering a culture of sustainability that acknowledges the reciprocal relationship between individual well-being and environmental health.

Moreover, our conclusions resonate with the shifting corporate landscape towards environmental, social, and governance (ESG) principles, emphasizing the synergistic potential of aligning business practices with sustainable and health-promoting consumer behaviors. This synergy underscores the importance of viewing health and environmental sustainability not as separate challenges but as intertwined elements of a holistic response to the contemporary crises facing our world.

In essence, this study offers a roadmap for embracing sports and leisure activities to foster sustainable consumerism, advocating for a future where businesses and consumers jointly contribute to a healthier, more sustainable planet. The journey towards sustainability transcends mere product choice, representing a broader commitment to personal health and the health of our environment, underscored by the foundational role of LTPA in this transformative process.

### 5.2. Theoretical Implications

This study contributes to the theoretical framework of sustainable business practices and environmental awareness by integrating concepts from time geography and leisure-time physical activity (LTPA). It reveals how the spatial and temporal dimensions of human activities intersect with the health and environmental benefits associated with LTPA, providing nuanced insights into consumer behavior towards green and sustainable products. Furthermore, it underscores the pivotal role of health and environmental awareness, influenced by leisure-time physical activity, in shaping consumers’ preferences for sustainable products. Recognizing this influential role is crucial for businesses and policymakers aiming to promote sustainability initiatives, highlighting the interconnectedness between individual behaviors, lifestyle choices, and environmental consciousness.

Firstly, the integration of time geography underscores the significance of considering the temporal and spatial context in which LTPA occurs [4]. This approach reveals how the allocation of discretionary time to physical activities not only enhances individual well-being but also fosters a deeper connection with the environment [6], thereby influencing eco-responsible consumer choices. The relationship between LTPA and reduced health risks [18,19], alongside its contribution to positive mental health and subjective well-being [20,21], exemplifies the intertwined nature of health and environmental consciousness.

Furthermore, this research advances the dialogue on green consumerism by emphasizing the role of LTPA in mediating awareness and attitudes towards sustainable products. It challenges the assumption that increased sports participation directly leads to green product consumption, suggesting instead that the awareness raised through LTPA acts as a critical mediator. This insight refines existing theories on green consumer behaviors, pointing to the need for holistic strategies that align health, environmental consciousness, and consumerism. Additionally, the findings affirm the relevance of ESG (environmental, social, and governance) principles in shaping business strategies for the green product sector. In this context, time geography and LTPA offer strategic avenues for businesses to engage consumers by highlighting the interconnectedness of health, environment, and sustainable living [64]. This approach not only demands authentic efforts from companies to promote sustainability but also encourages consumers to integrate eco-conscious values into their lifestyles, catalyzing transformative behaviors.

In sum, by weaving together time geography, LTPA, health awareness, and environmental consciousness, this study contributes a novel perspective to the discourse on sustainable consumerism and business strategies. It underscores the necessity of adopting integrated approaches that consider the physical and temporal dimensions of human activity, thereby enriching theoretical frameworks and guiding practical applications in the realm of sustainable business and green consumerism.

### 5.3. Managerial Implications

In the era of post-pandemic recovery, businesses face unprecedented challenges amidst environmental concerns and health crises exacerbated by the lasting impact of the COVID-19 pandemic. It is imperative for businesses to reevaluate their strategic approaches to adapt to the evolving landscape of consumer behavior. This study explores the intricate interplay between health consciousness, environmental awareness, and leisure sports engagement, aiming to provide actionable insights for businesses, particularly those operating within the green product sector. As consumers increasingly prioritize sustainable consumption in the wake of the pandemic, it becomes evident that such behaviors are not solely driven by participation in environmentally friendly or health-promoting activities. Rather, they are significantly influenced by the nuanced interaction between health and environmental awareness during the decision-making process of purchasing green and sustainable products. Given these findings, strategic recommendations for management are crucial in navigating the post-pandemic era effectively.

Leveraging Time Geography and LTPA Insights: Understanding the spatial and temporal dimensions of LTPA, businesses can develop targeted campaigns that underscore the interconnectedness of health, environmental awareness, and leisure activities. This approach can facilitate the creation of narratives that resonate with consumers’ lifestyles, highlighting the dual benefits of LTPA for personal well-being and environmental health.

Promoting Environmental and Health Consciousness: Beyond marketing products, companies should engage in initiatives that amplify the environmental and health benefits of LTPA. This could involve collaborations with community sports groups or outdoor education programs to encourage participation in leisure activities, fostering a deeper connection with nature and raising awareness of sustainability issues.

Aligning Marketing Strategies with ESG Goals: By communicating ESG commitments, businesses can leverage the synergies between LTPA, environmental consciousness, and sustainable living [6]. Marketing efforts should not only emphasize the eco-friendliness of products but also highlight how these products support a lifestyle that values health, outdoor activity, and environmental conservation.

Facilitating Outdoor Education and Collaboration: Collaborating with sports and leisure organizations to promote environmental education and outdoor activities can further reinforce the message of sustainability [65]. Supporting initiatives that educate consumers on how engaging with nature through leisure activities can lead to a more sustainable lifestyle can underscore the importance of preserving the environment for future recreational use.

By integrating these strategies, businesses can effectively navigate the post-pandemic landscape, recognizing that consumer purchasing decisions are increasingly influenced by a comprehensive awareness of health and environmental issues. This strategic adaptation not only positions businesses as leaders in sustainable development but also cultivates a consumer base that values and actively participates in preserving health and the environment through informed leisure choices.

### 5.4. Limitations and Further Research

This research, while providing insights into the dynamic interplay between engagement in sports and leisure activities and its influence on health and environmental consciousness, acknowledges certain constraints. Specifically, the study’s participant base, primarily from Taiwan, suggests a need for broader cultural and geographic exploration to ascertain the global applicability of our findings. Future studies should consider diverse populations to explore how cultural and regional variations affect the relationship between LTPA and green consumer behavior.

Moreover, the non-significance of certain hypotheses opens new directions for inquiry, particularly regarding whether specific types of LTPA have a more substantial impact on sustainable consumer choices. This question invites a more nuanced examination of how different physical activities influence environmental awareness and green purchasing decisions.

Additionally, the cross-sectional nature of this study limits our ability to observe the long-term effects of sports and leisure involvement on health and environmental consciousness. Longitudinal research could offer deeper insights into how sustained engagement with physical activities can foster lasting awareness and behavior change toward environmental sustainability.

Given these limitations, we advocate for future research to further explore the moderating roles of health and environmental consciousness within the context of LTPA. Employing qualitative methods, such as in-depth interviews or focus groups, could enrich quantitative findings and provide a more comprehensive understanding of the mechanisms driving green consumerism. This approach could uncover new pathways through which sports and leisure activities contribute to the cultivation of a sustainable mindset among consumers.

In sum, while this study sheds light on the critical roles of LTPA in promoting health and environmental awareness, it also underscores the complexity of translating these benefits into green consumer practices. Future research should aim to bridge these gaps, offering clearer guidance for leveraging sports and leisure as catalysts for sustainable living.

## Figures and Tables

**Figure 1 ijerph-21-00618-f001:**
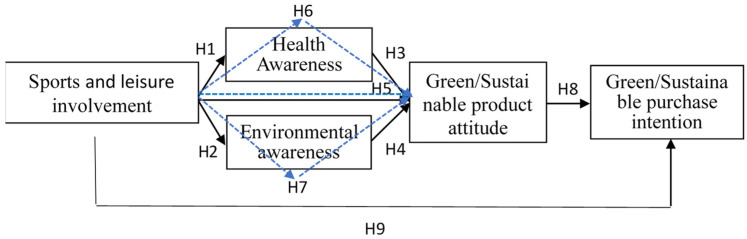
Research framework. Black arrows indicate direct effect influences (H1–H5, H8, H9). Blue arrows indicate mediating effect influences (H6–H7).

**Figure 2 ijerph-21-00618-f002:**
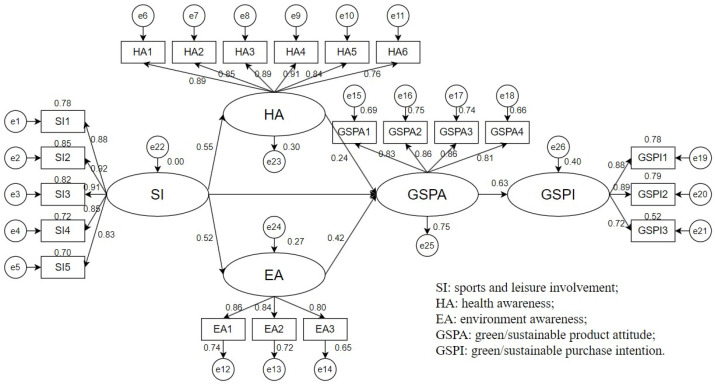
The model of structural equation modeling.

**Table 1 ijerph-21-00618-t001:** The results of reliability analysis.

Variable	Mean	SD	Cronbach’s α
Sports and leisure involvement	4.732	1.063	0.945
Health Awareness	4.887	0.948	0.942
Environmental Awareness	4.789	0.875	0.893
Green/Sustainable Product Attitude	4.655	0.925	0.915
Green/Sustainable Purchase Intention	4.122	1.029	0.892

**Table 2 ijerph-21-00618-t002:** The results of convergent validity analysis.

Concept	Item	Standard Factor Loading	Composite Reliability	Average Variance Extracted
Sports and leisure involvement (SI)	SI1	0.905	0.958	0.820
	SI2	0.930		
	SI3	0.925		
	SI4	0.888		
	SI5	0.878		
Health Awareness (HA)	HA1	0.902	0.954	0. 778
	HA2	0.873		
	HA3	0.905		
	HA4	0.917		
	HA5	0.874		
	HA6	0.815		
Environmental Awareness (EA)	EA1	0.906	0.934	0.824
	EA2	0.930		
	EA3	0.887		
Green/Sustainable Product Attitude (GSPA)	GSPA1	0.882	0.936	0.787
	GSPA2	0.903		
	GSPA3	0.898		
	GSPA4	0.865		
Green/Sustainable Purchase Intention (GSPI)	GSPI1	0.919	0.920	0.794
	GSPI2	0.921		
	GSPI3	0.830		

**Table 3 ijerph-21-00618-t003:** Discriminant validity.

Dimension	SI	HA	EA	GSPA	GSPI
SI	0.906 *				
HA	0.528	0.882 *			
EA	0.486	0.448	0.908 *		
GSPA	0.363	0.418	0.506	0.887 *	
GSPI	0.350	0.405	0.447	0.560	0.891 *

* The diagonal values are the square roots of AVEs.

**Table 4 ijerph-21-00618-t004:** Goodness-of-fit.

Goodness-of-Fit Index	Scholars	Estimate Required	Measurement Model
CMIN/DF	Joreskog and Sorbom [59]	<5	3.827
GFI	Doll, Xia and Torkzadeh [60]	>0.8	0.891
AGFI	MacCallum and Hong [61]	>0.8	0.860
NFI	Hair et al. [62]	>0.9	0.934
IFI	Hair et al. [62]	>0.9	0.950
CFI	Hair et al. [62]	>0.9	0.950
RMSEA	MacCallum and Hong [63]	<0.08	0.073

**Table 5 ijerph-21-00618-t005:** The results of SEM path analysis.

Hypothesis	Independent Variable	Dependent Variable	Regression Weights	S.E.	C.R.	*p*-Value
H1	SI	HA	0.464	0.037	12.522	***
H2	SI	EA	0.344	0.031	11.256	***
H3	HA	GSPA	0.200	0.043	4.661	***
H4	EA	GSPA	0.480	0.059	8.160	***
H5	SI	GSPA	0.031	0.041	3.987	0.454
H8	GSPA	GSPI	0.648	0.055	11.893	***
H9	SI	GSPI	0.146	0.036	4.019	***

*** *p* < 0.001.

**Table 6 ijerph-21-00618-t006:** Mediated effect analysis.

	Estimate	Confidence Interval 95%		Sig.
		*p*	BC	
Indirect effect				
Sports and leisure involvement → health awareness → Green/Sustainable Product Attitude	0.052	0.039	0.003~0.109	*
Sports and leisure involvement → Environmental Awareness → Green/Sustainable Product Attitude	0.113	0.001	0.061~0.176	*
Direct effect				
Sports and leisure involvement → Green/Sustainable Product Attitude	−0.015	0.812	−0.126~0.111	
Total effect				
Sports and leisure involvement → Green/Sustainable Product Attitude	0.151	0.002	0.049~0.262	*

* *p* < 0.001.

## Data Availability

The datasets used and/or analyzed during the current study are available from the first author on reasonable request at hjniu@mail.tku.edu.tw.

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
