# Peer review of "Leisure-Time Physical Activity as a Pathway to Sustainable Living: Insights on Health, Environment, and Green Consumerism"

_ijerph, 2024, doi:10.3390/ijerph21050618_

Round 1

Reviewer 1 Report

Comments and Suggestions for Authors

This study is an important topic that is relevant to current debates around wellbeing and environmental responsibility. The authors are commended for choosing this topic. Overall, the paper reads well and shows sufficient engagement with other scholarly work. There are, however, some improvements that can be made to the paper and I have made some suggestions below. 

Introduction

Well written, but a bit brief. 

The scope of the work can be better introduced. Is the study global? 

There are referencing issues in line 58 where they are inconsistent.

Line 61-what variables are being explored?

Literature review

Referencing error line 70.

Section 2.1 could be improved by making exclusive links of sports and leisure being conducted in natural spaces. Literature on wellbeing studies and natural healing gardens and mind body practices could assist to make this connection more apparent than it is currently.  The same applies for the review on health awareness. The review is being done outside of the connection to the environment.

H6 and H7 seem to be compounded linking one variable to two variables. Should they be stated as such or should they be two separate hypotheses each? 

Results

The results are quite substantial, with many of them feeding into the main test for mediation, which was H6 and H7. 

I seem to have missed the tests for the other hypotheses-H1 to 5. Were these tested? 

Discussion

The discussion is brief, and centered on H6 and H7. Not much attention is given to the others, how were the other dynamics shifting in your results. The methods should have nuanced some of the questions being asked as a representation of each of these variables. Whilst very good, they are also quite compound. Perhaps consider an appendix, or include some descriptive tables representing responses. to show how your questions were compounded and which represented each variable. This would help to open up your discussion a bit more.  

Regarding the methods, there is no description of the context, Taiwan, and also of the sampling procedure, other than age. Other variables such as wealth, and gender can affect responses, and this should be addressed somehow. I commented on the introduction, that this study presents as global, which needs to be addressed. Other studies in different contexts with different cultures may present different findings. Methodology is also misspelt.

The conclusion addressing consumerism is good, but perhaps limited to results of H7 and H6, which can be addressed should the authors wish to add more results for the other hypotheses.

Author Response

1.This study is an important topic that is relevant to current debates around wellbeing and environmental responsibility. The authors are commended for choosing this topic. Overall, the paper reads well and shows sufficient engagement with other scholarly work. There are, however, some improvements that can be made to the paper and I have made some suggestions below. 
=>Thank you for acknowledging the importance of our study's topic and the relevance it holds in current debates on wellbeing and environmental responsibility. We appreciate your positive remarks on how the paper reads and our engagement with scholarly work. We are also grateful for your constructive suggestions for improvement and will carefully consider each to enhance our paper.

 2.Introduction Well written, but a bit brief. The scope of the work can be better introduced. Is the study global? 
=>Thank you for your feedback. We appreciate your observation regarding the scope of our study. While our research delves into global issues such as environmental impact and the COVID-19 pandemic, it is important to note that our investigation is specifically focused on Taiwan. In our revised manuscript, we will provide a clearer introduction to the scope of our work, highlighting its relevance within the context of Taiwan while acknowledging its broader implications within the global landscape. We hope this clarification will address your concerns and provide a more comprehensive understanding of our study. Additionally, we greatly appreciate your recognition of the clarity of our writing. Following your suggestions, we are committed to revising the manuscript to enrich its content.

3.There are referencing issues in line 58 where they are inconsistent.
=>Thank you for your commendation, and we have revised.

 4.Line 61-what variables are being explored?
=>Thank you for your inquiry. The variables explored in our study encompass sports and leisure involvement, health awareness, environment awareness, green/sustainable product attitude, and purchase intention(Line88-90). These variables are pivotal in understanding consumer behavior towards green and sustainable products and are thoroughly examined throughout our research.

Literature review
5.Referencing error line 70.
=>Thank you for your commendation, and we have revised.

6.Section 2.1 could be improved by making exclusive links of sports and leisure being conducted in natural spaces. Literature on wellbeing studies and natural healing gardens and mind body practices could assist to make this connection more apparent than it is currently.  The same applies for the review on health awareness. The review is being done outside of the connection to the environment.
=>Thank you for your insightful suggestions regarding Section 2.1 of our manuscript. We appreciate your feedback on the need to strengthen the connection between sports and leisure activities and natural spaces. In response to your recommendation, we have incorporated references to literature on wellbeing studies, natural healing gardens, and mind-body practices to elucidate this connection more comprehensively. Additionally, we have revisited the review on health awareness and ensured that it is more explicitly linked to the environment, as per your suggestion. We believe these modifications have enhanced the clarity and relevance of our discussion, and we thank you for guiding us in refining our manuscript.

 7.H6 and H7 seem to be compounded linking one variable to two variables. Should they be stated as such or should they be two separate hypotheses each? 
=>Thank you for your inquiry regarding the conceptualization of mediating variables. Mediating variables serve as intermediaries in the relationship between an independent variable and a dependent variable. They help explain the underlying mechanism or process through which the independent variable influences the dependent variable. In the context of H6 and H7, if one variable is purported to influence two other variables, it suggests a potential mediating relationship. Rather than treating them as separate hypotheses, framing them as one hypothesis with a mediating variable can provide a more comprehensive understanding of the underlying dynamics. This approach allows for the examination of how the mediating variable mediates the relationship between the independent variable and each of the dependent variables. By considering H6 and H7 in this light, we aim to elucidate the mediating role of the variable in question and provide insights into the underlying processes driving the relationships among the variables of interest. This approach aligns with the principles of mediation analysis, which seeks to uncover the mechanisms through which variables exert their effects on one another.

8.The methods should have nuanced some of the questions being asked as a representation of each of these variables. Whilst very good, they are also quite compound. Perhaps consider an appendix, or include some descriptive tables representing responses. to show how your questions were compounded and which represented each variable. This would help to open up your discussion a bit more.  

Regarding the methods, there is no description of the context, Taiwan, and also of the sampling procedure, other than age. Other variables such as wealth, and gender can affect responses, and this should be addressed somehow. I commented on the introduction, that this study presents as global, which needs to be addressed. Other studies in different contexts with different cultures may present different findings. Methodology is also misspelt.
=>Thank you for your valuable insights into the methods section of our study. We appreciate your feedback regarding the need for more detailed information about Taiwan and the sampling procedure beyond age. In our revised manuscript, we will ensure to address these concerns by providing a clearer explanation of the scope of our study, including a more thorough description of the context, sampling procedure, and addressing any misspelled terms. Your suggestions are greatly appreciated, and we will incorporate them into our revision accordingly.

Results
9.The results are quite substantial, with many of them feeding into the main test for mediation, which was H6 and H7. 
I seem to have missed the tests for the other hypotheses-H1 to 5. Were these tested? 
=>Thank you for your detailed comments and suggestions. Indeed, the results of our study are rich, especially the mediating effects (H6 and H7), which are most crucially and closely related to the entire research. Regarding the other hypotheses (H1 to H5) you mentioned, we have indeed conducted thorough analyses and tests. In response to your suggestions, we have reviewed the related data and analytical methods, and included the results of these hypotheses in the discussion section. Additionally, we have rewritten the conclusion section to present our research findings and related conclusions more clearly. We hope these revisions will better meet the review standards and clearly convey our research contributions.

Discussion
10.The discussion is brief and centered on H6 and H7. Not much attention is given to the others, how were the other dynamics shifting in your results.
The conclusion addressing consumerism is good, but perhaps limited to results of H7 and H6, which can be addressed should the authors wish to add more results for the other hypotheses.
=>We are pleased to hear that you found the conclusion addressing consumerism satisfactory. We appreciate your observation regarding the focus of the discussion on H6 and H7. We understand the importance of giving equal attention to all hypotheses and their respective dynamics in our results. To rectify this imbalance, we will ensure that the discussion thoroughly examines each hypothesis. Furthermore, we have rewritten the discussion section to present our research findings and related conclusions more clearly. We believe these revisions will better meet the review standards and effectively convey our research contributions.

Reviewer 2 Report

Comments and Suggestions for Authors

The investigation is sound and the paper is very well written. The authors take care about the methods and are very cautious and systematic in the way they present the results. The authors are very didactic and explain very well what they do and how they provide the results.

Although the research is not rocket science, the results are interesting and well justified. The use of leasure time to do physical activity and the relation of this with the patterns of consumption is sound and very timely, and it very well deserves a piece of research as the one presented in this manuscript. 

One minnor detail about the English:

- Section 2.3, second paragraph: On the one hand... 

Author Response

  1. The investigation is sound and the paper is very well written. The authors take care about the methods and are very cautious and systematic in the way they present the results. The authors are very didactic and explain very well what they do and how they provide the results.
    =>Thank you for your positive feedback and recognition of our systematic approach and clear presentation. We appreciate your encouraging comments.
  1. Although the research is not rocket science, the results are interesting and well justified. The use of leasure time to do physical activity and the relation of this with the patterns of consumption is sound and very timely, and it very well deserves a piece of research as the one presented in this manuscript. One minnor detail about the English:- Section 2.3, second paragraph: On the one hand... 
    =>Thank you for your commendation, and we appreciate your positive feedback on our manuscript. We have taken note of your minor detail regarding the writing in Section 2.3, second paragraph, where "On the one hand..." is mentioned. We will ensure to review and address this linguistic aspect to maintain clarity and coherence throughout the manuscript.

Reviewer 3 Report

Comments and Suggestions for Authors

I really appreciate the discussion of an important and timely topic.
In line 58 and line 70, remove the names of the authors of the articles you refer to. That's why there are footnotes in the form of numbers.
Hypothesis H1 seems obvious to me, as is hypothesis H4. In my opinion, they do not require empirical confirmation. I suggest changing or removing them - you proposed a large number of hypotheses for a scientific article.
Moreover, not all hypotheses are directly verified in the conclusions. This needs to be completed.
There is a cheat sheet in line 195, and instead of Methodology, I suggest: Material and Methods. Methodology is the science that deals with methods.

You adopted a 6-point Likert scale. This is a very rare approach. It is usually 5 or 7 degrees. This should be better justified and at the same time it should be stated what answers each number on the scale corresponds to (all numbers from 1 to 6).

There is no at least a short discussion of the results.

I appreciate the description of the limitations of this study.

Author Response

1.I really appreciate the discussion of an important and timely topic.
=>Thank you for your commendation, and we appreciate your positive feedback on our manuscript.

2.In line 58 and line 70, remove the names of the authors of the articles you refer to. That's why there are footnotes in the form of numbers.
=>Thank you for your commendation, and we have revised.

3.Hypothesis H1 seems obvious to me, as is hypothesis H4. In my opinion, they do not require empirical confirmation. I suggest changing or removing them - you proposed a large number of hypotheses for a scientific article.Moreover, not all hypotheses are directly verified in the conclusions. This needs to be completed.
=>Thank you for your feedback and for sharing your perspective on hypotheses H1 and H4. We understand your point about the perceived obviousness of these hypotheses. However, it is essential to include these foundational hypotheses to ensure the overall integrity and completeness of the research. These hypotheses serve as fundamental building blocks on which subsequent analyses and conclusions are based. Therefore, we believe it is prudent to retain them in our study.

Regarding your observation that not all hypotheses are directly verified in the conclusions, we agree that this aspect needs improvement for a more comprehensive discussion. We have rewritten the conclusions section to ensure that all hypotheses are appropriately discussed, and their implications are adequately addressed. This adjustment will contribute to a more thorough and coherent conclusion that aligns with the objectives and findings of the study.

4.There is a cheat sheet in line 195, and instead of Methodology, I suggest: Material and Methods. Methodology is the science that deals with methods.
=>Thank you for your suggestion regarding the terminology on line 195, which should indeed be line 295. We have revised it to "Materials and Methods" as per your recommendation, which is indeed more precise. We appreciate your attention to detail.

5.You adopted a 6-point Likert scale. This is a very rare approach. It is usually 5 or 7 degrees. This should be better justified and at the same time it should be stated what answers each number on the scale corresponds to (all numbers from 1 to 6).
=>Thank you for your insightful feedback regarding the Likert scale used in our study. We appreciate your attention to detail in this matter. As you noted, we employed a 6-point Likert scale, which may seem uncommon compared to the more typical 5 or 7-point scales. However, our decision to use a 6-point scale was based on research suggesting its effectiveness in measuring personal attributes (Chomeya, 2010).

“ Chomeya (2010) describes the 4-Point and 6-Point Likert Type Scales as appropriate models for measurement, providing respondents with clear options for their answers. By utilizing a 6-point scale, we aimed to offer respondents a sufficient range of options while maintaining clarity and ease of interpretation. Each number on the scale corresponds to a specific response option, ranging from strongly disagree to strongly agree. We will explicitly state this correspondence in the methodology section of our manuscript to ensure clarity for readers.”

Overall, we believe that using a 6-point Likert scale enhances the precision and reliability of our data collection process, aligning with established models and providing respondents with a structured framework for their responses.

Chomeya, R. (2010). Quality of psychology test between Likert scale 5 and 6 points. Journal of Social

6.There is no at least a short discussion of the results.

=>Thank you for your valuable feedback regarding the discussion section of our manuscript. We are grateful for your guidance and have revised the manuscript accordingly. We are optimistic that these amendments will meet the journal's standards. Thank you again for helping us improve our work.

7. I appreciate the description of the limitations of this study.
=>Thank you for your praise, as it encourages continual improvement and critical thinking within our work. Thank you once again for your constructive feedback.